# Does the skull Hounsfield unit predict shunt dependent hydrocephalus after decompressive craniectomy for traumatic acute subdural hematoma?

In-Suk Bae[1], Jae Min Kim[2], Jin Hwan Cheong[2], Je Il Ryu[2], Kyu-Sun Choi[3], Myung-Hoon Han[2]*

1 Department of Neurosurgery, Eulji University Eulji Hospital, Seoul, Korea, 2 Department of Neurosurgery, Hanyang University Guri Hospital, Gyonggi-do, Korea, 3 Department of Neurosurgery, Hanyang University Medical Center, Seoul, Korea

* gksmh80@gmail.com

**Data Availability Statement:** All relevant data are within the paper and its Supporting Information files.

## Abstract

### Background and purpose

Posttraumatic hydrocephalus affects 11.9%–36% of patients undergoing decompressive craniectomy (DC) after traumatic brain injury and necessitates a ventriculo-peritoneal shunt placement. As bone and arachnoid trabeculae share the same collagen type, we investigated possible connections between the skull Hounsfield unit (HU) values and shunt-dependent hydrocephalus (SDHC) in patients that received cranioplasty after DC for traumatic acute subdural hematoma (SDH).

### Methods

We measured HU values in the frontal bone and internal occipital protuberance from admission brain CT. Receiver operating characteristic curve analysis was performed to identify the optimal cut-off skull HU values for predicting SDHC in patients receiving cranioplasty after DC due to traumatic acute SDH. We investigated independent predictive factors for SDHC occurrence using multivariable logistic regression analysis.

### Results

A total of 162 patients (>15 years of age) were enrolled in the study over an 11-year period from two university hospitals. Multivariable logistic analysis revealed that the group with simultaneous frontal skull HU $\leq$797.4 and internal occipital protuberance HU $\leq$586.5 (odds ratio, 8.57; 95% CI, 3.05 to 24.10; P<0.001) was the only independent predictive factor for SDHC in patients who received cranioplasty after DC for traumatic acute SDH.

### Conclusions

Our study reveals a potential relationship between possible low bone mineral density and development of SDHC in traumatic acute SDH patients who had undergone DC. Our

**Funding:** This work was supported by the research fund of Hanyang University (HY-201900000003370).

**Competing interests:** The authors have declared that no competing interests exist.

findings provide deeper insight into the association between low bone mineral density and hydrocephalus after DC for traumatic acute SDH.

## Introduction

Acute traumatic subdural hematoma (SDH) is a common medical emergency among patients experiencing traumatic brain injury (TBI) [1]. Increasingly, decompressive craniectomy (DC) is performed to control medically refractory intracranial hypertension in patients with TBI [2,3]. The increase in number of DC surgeries has led to an increase in subsequent cranioplasty (CP) procedures [4]. Posttraumatic hydrocephalus affects 11.9%–36% of patients undergoing DC after traumatic brain injury (TBI) [5]. Approximately 27% of patients undergoing CP require a ventriculo-peritoneal (V-P) shunt placement [6]. To correct posttraumatic hydrocephalus for good outcomes for patients with DC, simultaneous or staged V-P shunt is necessary with or following cranioplasty [7]. The cause of development of posttraumatic hydrocephalus after DC is considered multi-factorial, and risk factors associated with shunt-dependent hydrocephalus (SDHC) remain unclear.

To the best of our knowledge, there have been no studies investigating the association between skull Hounsfield unit (HU) values from CT scans and development of SDHC in patients who had undergone DC for traumatic acute SDH. We initiated this study based on the premise that both the arachnoid trabeculae and granulations are composed of type 1 collagen, which is also the principal component of bone matrix protein [8]. Therefore, we hypothesized that low bone mineral density (BMD) would negatively influence the integrity of the arachnoid trabeculae and granulations as well as bone. We conjectured that this mechanism may be associated with development of SDHC after DC for traumatic SDH.

To assess this hypothesis, we measured HU values in the frontal bone and internal occipital protuberance (IOP) from admission brain CT in patients who had undergone DC for traumatic acute SDH. Further, we examined the relationship between skull HU values and development of SDHC after traumatic SDH.

## Methods

### Study design

We retrospectively extracted data for acute traumatic SDH patients older than 15 years of age from the database of TBI patients consecutively admitted to the department of neurosurgery of Hanyang University Medical Center (Seoul and Guri) in the period between January 1, 2007, and December 31, 2017. Initially, many patients with acute traumatic SDH who underwent DC had died or were lost to follow-up for various reasons. Therefore, to reduce heterogeneity of study populations and assess risk factors for ultimately needing V-P shunt among SDH patients who had undergone DC, we excluded patients who did not undergo cranioplasty. To identify risk factors for SDHC in patients receiving cranioplasty after craniectomy due to traumatic subdural hematoma, we included only patients who met the following study-specific criteria: (1) received unilateral or bilateral DC due to acute traumatic SDH, and (2) ultimately underwent cranioplasty with or without V-P shunts. Among the patients who met the inclusion criteria, we excluded patients with (1) surgery performed more than 48 hours after head trauma, (2) history of previous brain surgery, brain tumor, or stroke (ischemic or hemorrhagic), and (3) no measurable cancellous bone of the frontal skull in brain CT. The

remaining 162 patients were finally included in the study. All study patients were well followed up after DC and underwent subsequent cranioplasty; there were no missing data.

This study was approved by the Institutional Review Boards of Hanyang University Medical Center in both Seoul and Guri and conformed with the tenets of the Declaration of Helsinki. Owing to the retrospective nature of the study, the requirement for informed consent was waived. All individual records were anonymized prior to analysis.

### Surgery and management

DC was performed in a standardized manner using a trauma flap with expansive duroplasty [9]. Surgical techniques and patient management were similar across the neurosurgeons in both hospitals as they trained in the same hospital (Hanyang University Medical Center). All SDH patients undergoing DC were managed in the intensive care unit according to a standard protocol including mechanical ventilation, intracranial pressure control with mannitol, anti-convulsant agent, and glucose level control with insulin [10]. Cranioplasty was performed after resolution of cerebral edema on CT scans using mostly autologous bone, in a standardized manner as previously reported elsewhere [11]. The timing of cranioplasty varied according to the surgeon's preference and was not strictly standardized due to the retrospective nature of the study. V-P shunt was performed when the patients had hydrocephalus-related symptoms with the presence of ventricular enlargement on brain CT scans. Hydrocephalus-related symptoms for considering V-P shunt were defined as neurological deterioration and aggravated response to painful stimuli, especially in stuporous patients, and additional decline in the daily functions, gait disturbance, urinary incontinence, or confusion, especially in conscious patients. Simultaneous or staged V-P shunting was performed with or following cranioplasty.

### Clinical and radiographic variables

We investigated possible associated factors for the development of SDHC in patients receiving cranioplasty after DC for traumatic acute SDH. All clinical information of the enrolled patients was investigated by two trained research members using electronic medical records. Clinical data including sex, age, body mass index (BMI), side of craniectomy, reoperation, Glasgow coma scale (GCS), hypertension, and diabetes were investigated from medical and operative records. BMI was calculated as weight/(height × height) and expressed in kg/m$^2$. Visual, verbal, and motor components of the GCS score were recorded for each patient at the time of initial trauma.

We analyzed the initial CT images obtained in the emergency room for all study patients. Radiographic variables including degree of midline shift, accompanying traumatic subarachnoid hemorrhage (SAH), intracerebral hemorrhage (ICH), intraventricular hemorrhage (IVH), epidural hematoma (EDH), and skull fracture were evaluated and confirmed by two faculty neurosurgeons blinded to the clinical data using the picture archiving and communication system (PACS).

### Measurement of frontal skull and IOP HU

All CT images (4.0–5.0-mm slice thicknesses) were obtained with Philips and Siemens CT scanners in both hospitals. Birnbaum BA et al. reported that variations in HU values are very small (range of 0–20 HU) between five CT scanners including Philips and Siemens [12]. We previously reported detailed methods for measuring HU values at each of four lines on the frontal cancellous bone between the right and left coronal sutures on axial CT slices at the point where the lateral ventricles disappear (Fig 1A) [13,14].

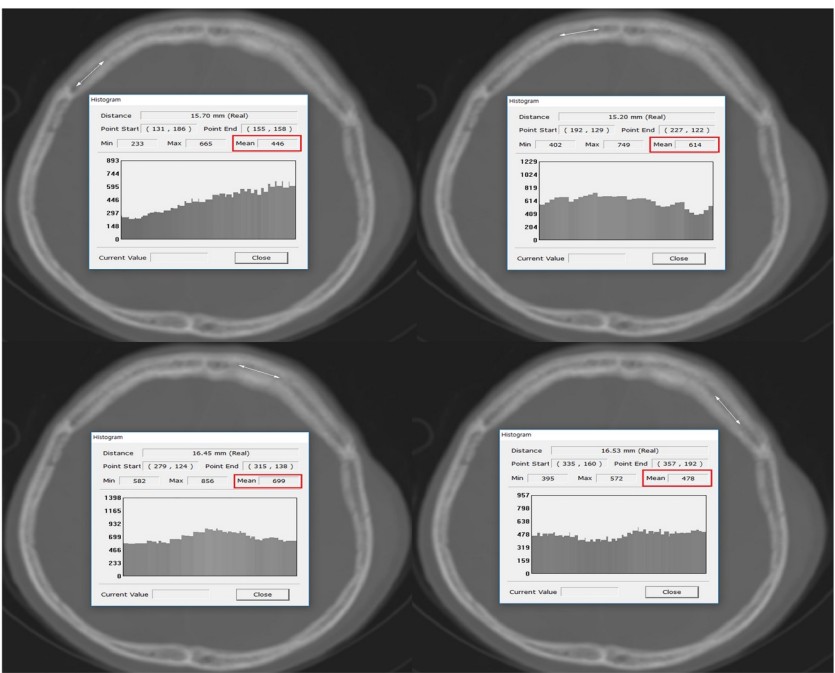

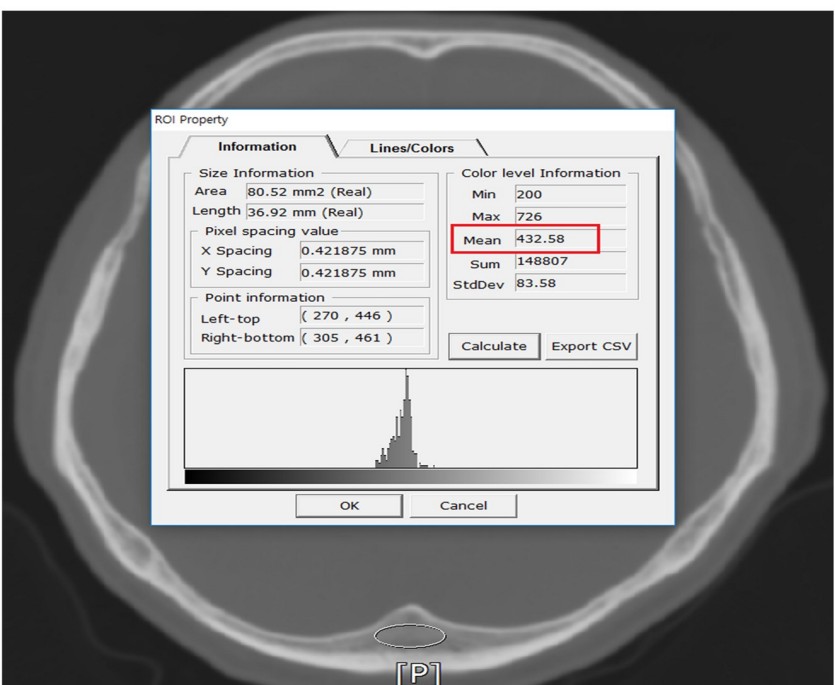

**Fig 1. Measurement of HU values at frontal bone and IOP.** (A) Measurement of HU values at each of four lines on the frontal bone (right lateral, right medial, left medial, and left lateral). The PACS automatically calculated the maximum, minimum, and mean HU values according to the values on the drawn line, and the mean HU value on each of the four lines was recorded. (B) Measurement of HU values at IOP. The PACS automatically calculated the maximum, minimum, and mean HU values according to the oval ROI. The mean HU value was used for the study. HU = Hounsfield unit; IOP = internal occipital protuberance; PACS = picture archiving and communication system; ROI = region of interest.

The HU value of cancellous bone at the IOP of the occipital bone was measured using the oval "region of interest (ROI)" function in the PACS (Fig 1B). The PACS automatically calculated the maximum, minimum, and mean HU values according to the values of the ROI. We recorded the mean HU value of ROI of cancellous bone at the IOP. To reduce measurement errors, all brain CT images were magnified for HU measurement. All frontal skull and IOP HU measurements were conducted by a faculty neurosurgeon blinded to the clinical data of all patients.

## Statistical methods

Continuous variables are expressed as mean ± SD or median with interquartile range. Discrete variables are expressed as counts and percentage. The chi-square test and Student's t-test were used to assess differences between non-SDHC and SDHC groups. The mean frontal skull HU ([mean right lateral HU + mean right medial HU + mean left medial HU + mean left lateral HU]/4) and mean IOP HU values were used in all analyses.

We constructed scatterplots with regression lines to represent the associations between mean frontal skull HU and IOP HU values. Box plots with dot plots were used to visualize the association between the mean frontal skull HU and IOP HU values and SDHC.

Receiver operating characteristic (ROC) curve analysis was performed to identify the optimal cut-off values of frontal skull and IOP HU for predicting SDHC in patients receiving cranioplasty after craniectomy due to traumatic subdural hematoma. The HU value with the maximum concurrent sensitivity and specificity was considered the optimal cutoff value.

Odds ratios (ORs) with 95% confidence intervals (CIs) were estimated using uni- and multivariable logistic regressions to determine the independent predictive factors for the development of SDHC in patients that received cranioplasty after DC for acute SDH. Sex, age (continuous variable), BMI (continuous variable), patients with simultaneous frontal skull HU ≤797.4 and IOP HU ≤586.5, side of craniectomy, reoperation, midline shifting, initial GCS (continuous variable), traumatic SAH, ICH, IVH, EDH, skull fracture, hypertension, and diabetes were entered into the multivariable model. The Hosmer-Lemeshow goodness of fit test was used to analyze the fit of the logistic regression model. Additional interaction analysis was performed using the R package, "effects".

P-values less than 0.05 were considered statistically significant. All statistical analyses were performed using R software version 3.5.2 and SPSS for Windows, version 24.0 software (IBM, Chicago, IL).

## Results

### Characteristics of study patients

A final total of 162 patients (>15 years of age) who had undergone cranioplasty after DC for traumatic acute SDH with no previous history of brain surgery, brain tumor, or stroke over an 11-year period from two university hospitals were enrolled in the study. Mean patient age was 57.4 years, and 29.0% of patients were female. In total, 33 patients (20.4%) were treated with V-P shunts for hydrocephalus in patients receiving cranioplasty after DC for acute SDH. In total, 146 patients underwent unilateral craniectomy for traumatic acute SDH. Among 16 patients with bilateral DC, five patients underwent bilateral DC separately due to newly developed contralateral hemorrhagic lesions, and 11 patients underwent a single procedure. Overall, 18 patients (11.1%) had surgery related complications requiring reoperation. This study only included patients ultimately receiving cranioplasty after DC for traumatic acute SDH and indicated relatively lower postoperative complications requiring reoperation compared to those of

**Table 1. Clinical characteristics of patients ultimately receiving cranioplasty after craniectomy due to traumatic subdural hematoma classified according to shunt-dependent hydrocephalus in the study cohort.**

| Characteristics | SDHC (-) | SDHC (+) | Total | P |
|---|---|---|---|---|
| Number (%) | 129 (79.6) | 33 (20.4) | 162 (100) | |
| Sex, female, n (%) | 37 (28.7) | 10 (30.3) | 47 (29.0) | 1.000 |
| Age, mean ± SD, y | 56.9 ± 18.0 | 59.1 ± 13.5 | 57.4 ± 17.1 | 0.520 |
| BMI, mean ± SD, kg/m$^2$ | 23.2 ± 4.4 | 23.3 ± 3.4 | 23.2 ± 4.2 | 0.879 |
| Height, mean ± SD, m | 1.6 ± 0.1 | 1.7 ± 0.1 | 1.6 ± 0.1 | |
| Weight, mean ± SD, kg | 63.5 ± 14.7 | 64.0 ± 10.5 | 63.6 ± 13.9 | |
| Side of craniectomy, n (%) | | | | 0.088 |
| Right | 58 (45.0) | 17 (51.5) | 75 (46.3) | |
| Left | 61 (47.3) | 10 (30.3) | 71 (43.8) | |
| Bilateral | 10 (7.8) | 6 (18.2) | 16 (9.9) | |
| Reoperation, n (%) | 14 (10.9) | 4 (12.1) | 18 (11.1) | 0.521 |
| Postoperative complications | 6 (4.7) | 2 (6.1) | 8 (4.9) | |
| Infections | 8 (6.2) | 2 (6.1) | 10 (6.2) | |
| Midline shift on admission, mean ± SD, mm | 11.9 ± 3.9 | 13.5 ± 4.5 | 12.2 ± 4.1 | 0.043 |
| ≤ 10, n (%) | 31 (24.0) | 5 (15.2) | 36 (22.2) | |
| > 10 and ≤ 20, n (%) | 96 (74.4) | 26 (78.8) | 122 (75.3) | |
| > 20, n (%) | 2 (1.6) | 2 (6.1) | 4 (2.5) | |
| Initial Glasgow coma scale, median (IQR) | 9 (9–12) | 9 (9–10) | 9 (9–11.3) | 0.274 |
| Accompanying injury, n (%) | | | | |
| SAH | 96 (74.4) | 23 (69.7) | 119 (73.5) | 0.743 |
| ICH | 77 (59.7) | 20 (60.6) | 97 (59.9) | 1.000 |
| IVH | 25 (19.4) | 6 (18.2) | 31 (19.1) | 1.000 |
| EDH | 23 (17.8) | 6 (18.2) | 29 (17.9) | 1.000 |
| Skull fracture | 39 (30.2) | 8 (24.2) | 47 (29.0) | 0.644 |
| Past medical history, n (%) | | | | |
| Hypertension | 41 (31.8) | 9 (27.3) | 50 (30.9) | 0.772 |
| Diabetes | 25 (19.4) | 3 (9.1) | 28 (17.3) | 0.256 |

SDHC, shunt-dependent hydrocephalus; SD, standard deviation; BMI, body mass index; IQR, interquartile range; SAH, subarachnoid hemorrhage; ICH, intracerebral hemorrhage; IVH, intraventricular hemorrhage; EDH, epidural hematoma

a recent study [15]. There was a significant difference in midline shift on admission between patients with SDHC and those without. Further descriptive data are shown in Table 1.

## Association between mean frontal skull HU and IOP HU values

Fig 2 depicts a positive correlation between mean frontal skull HU and IOP HU values.

We observed an approximate increase of 0.8 HU of IOP per 1 HU increase of mean frontal skull HU with approximately 70% explanatory power (B, 0.777; P<0.001, $R^2$ = 0.699).

## Skull HU values according to SDHC in study patients

Table 2 contains descriptive statistics of frontal skull and IOP HU values according to SDHC.

We observed significant differences in values of mean frontal skull HU, IOP HU, and classification of skull HU between non-SDHC and SDHC groups. The overall average mean frontal skull HU value was 797.4 in all study patients, and 825.4 in the non-SDHC group and 687.8 in the SDHC group. Similarly, the overall mean IOP HU value was 601.4 in all study patients,

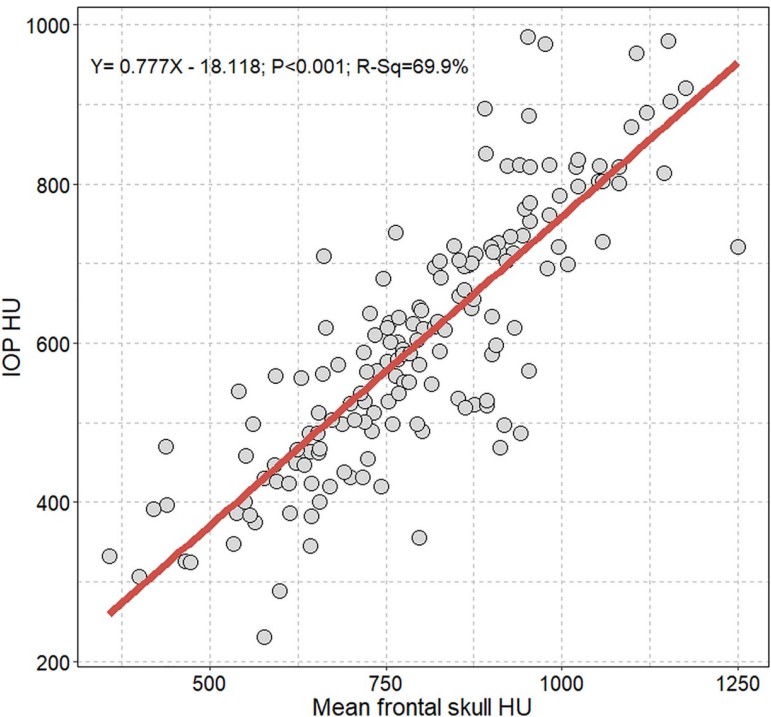

**Fig 2. Scatterplot with linear regression line showing the association between mean frontal skull HU and IOP HU values.** HU = Hounsfield unit; IOP = internal occipital protuberance.

and 625.5 in the non-SDHC group and 507.3 in the SDHC group. The boxplot graphically depicts significantly lower mean frontal skull and IOP HU values in the SDHC group than in the non-SDHC group (Fig 3A and 3B).

**Table 2. Descriptive statistics of frontal and IOP HU values in the study cohort classified according to shunt-dependent hydrocephalus.**

| Characteristics | SDHC (-) | SDHC (+) | Total | P |
|---|---|---|---|---|
| Overall mean frontal skull HU value, median (IQR) | 824.0 (719.3–940.8) | 681.8 (586.1–786.0) | 794.8 (669.1–914.1) | < 0.001 |
| Overall mean frontal skull HU value, mean ± SD | 825.4 ± 167.9 | 687.8 ± 141.7 | 797.4 ± 171.7 | < 0.001 |
| Mean HU value at each of four sites in the frontal skull, mean ± SD | | | | |
| Right lateral | 778.5 ± 162.5 | 665.4 ± 147.6 | 755.4 ± 165.6 | < 0.001 |
| Right medial | 861.8 ± 193.8 | 707.9 ± 155.1 | 830.4 ± 196.2 | < 0.001 |
| Left medial | 860.9 ± 180.2 | 730.0 ± 148.8 | 834.2 ± 181.7 | < 0.001 |
| Left lateral | 800.3 ± 160.7 | 647.9 ± 169.2 | 769.3 ± 173.2 | < 0.001 |
| Average, medial | 861.3 ± 183.0 | 718.9 ± 147.2 | 832.3 ± 185.0 | < 0.001 |
| Average, lateral | 789.4 ± 158.0 | 656.7 ± 149.0 | 762.4 ± 164.7 | < 0.001 |
| Overall IOP HU value, median (IQR) | 620.0 (499.5–726.5) | 504.0 (399.0–579.5) | 587.5 (487.0–713.5) | < 0.001 |
| Overall IOP HU value, mean ± SD | 625.5 ± 154.2 | 507.3 ± 146.9 | 601.4 ± 159.6 | < 0.001 |
| Classification of skull HU | | | | |
| Mean frontal skull HU ≤797.4 | 57 (44.2) | 27 (81.8) | 84 (51.9) | < 0.001 |
| IOP HU ≤586.5 | 54 (41.9) | 26 (78.8) | 80 (49.4) | < 0.001 |
| Simultaneous frontal skull HU ≤797.4 and IOP HU ≤586.5, n (%) | 43 (33.3) | 25 (75.8) | 68 (42.0) | < 0.001 |

SDHC, shunt-dependent hydrocephalus; HU, Hounsfield unit; IQR, interquartile range; SD, standard deviation; IOP, internal occipital protuberance

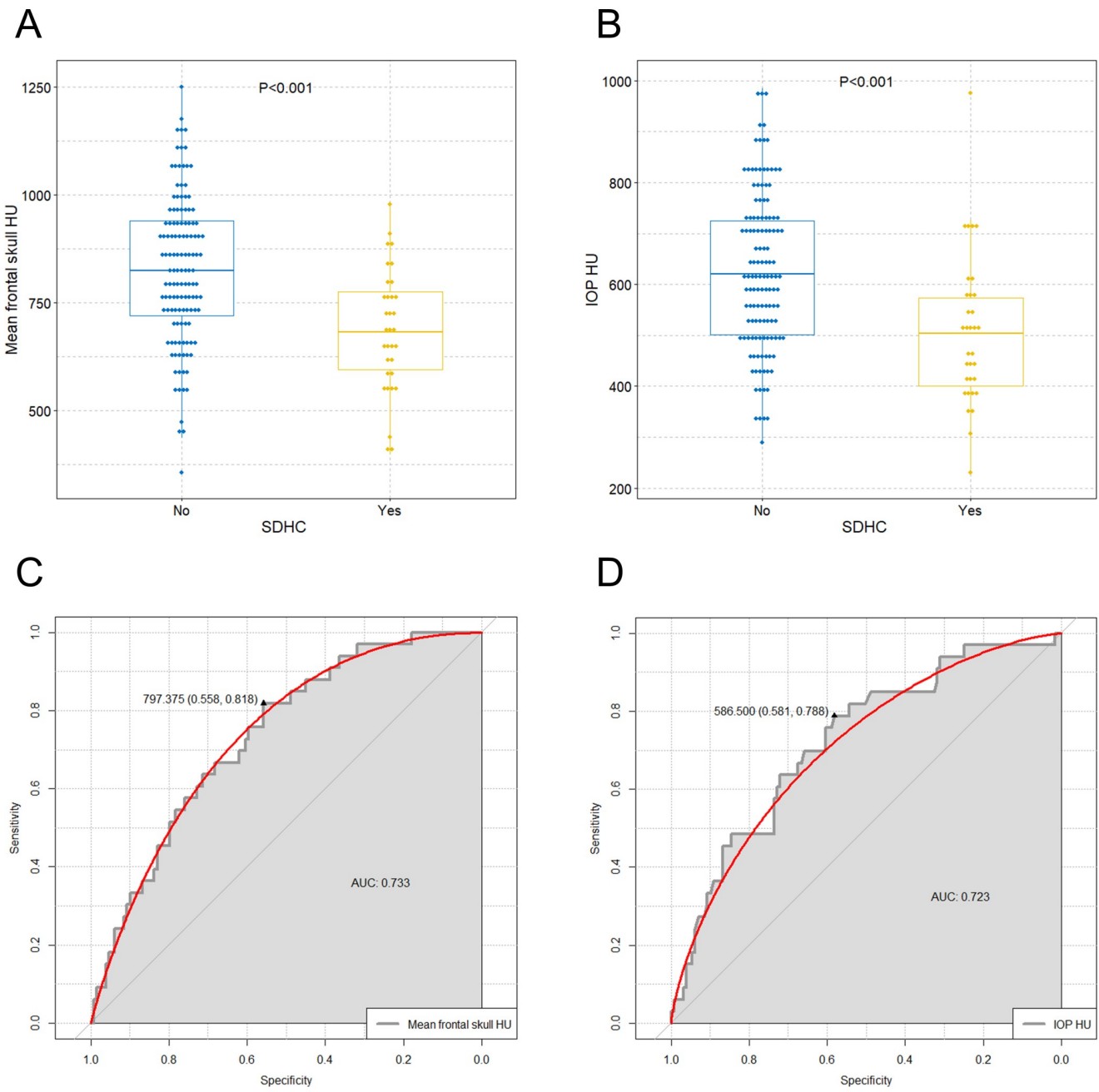

**Fig 3. Comparison of mean frontal skull HU and IOP HU values between SDHC and non-SDHC groups and determination of the optimal cut-off skull HU values for prediction of SDHC after DC for traumatic acute SDH.** Boxplots with dot plots of (A) mean frontal skull HU values and (B) IOP HU values according to SDHC. ROC curve to identify the optimal cutoff values of (C) mean frontal skull HU values and (D) IOP HU values for prediction of SDHC. HU = Hounsfield unit; IOP = internal occipital protuberance; SDHC = shunt-dependent hydrocephalus; DC = decompressive craniectomy; SDH = subdural hematoma; ROC = receiver operating characteristic; AUC = area under the curve.

## Determination of the optimal cut-off values of frontal and occipital skull HU for prediction of SDHC

The optimal cut-off values of the mean frontal skull HU and IOP HU for prediction of SDHC in patients receiving cranioplasty after DC for acute SDH were 797.375 (area under the curve [AUC] = 0.733; sensitivity = 81.8%; specificity = 55.8%; P<0.001) and 586.500 (AUC = 0.723;

**Table 3. Univariate logistic regression analysis of shunt-dependent hydrocephalus in patients ultimately receiving cranioplasty after craniectomy due to traumatic subdural hematoma based on various predictive factors.**

| Variable | Univariate logistic regression analysis | | |
| --- | --- | --- | --- |
| | OR | 95%CI | P |
| Sex | | | |
| Female (vs male) | 1.08 | 0.47 to 2.49 | 0.855 |
| Age (per 1-year increase) | 1.01 | 0.99 to 1.03 | 0.517 |
| BMI (per 1 BMI increase) | 1.01 | 0.92 to 1.10 | 0.878 |
| Classification of skull HU | | | |
| Mean frontal skull HU ≤797.4 (vs the other) | 5.68 | 2.20 to 14.70 | <0.001 |
| IOP HU ≤586.5 (vs the other) | 5.16 | 2.09 to 12.75 | <0.001 |
| Simultaneous frontal skull HU ≤797.4 and IOP HU ≤586.5 (vs other groups) | 6.25 | 2.60 to 15.01 | <0.001 |
| Side of craniectomy | | | |
| Left (vs right) | 0.56 | 0.24 to 1.32 | 0.185 |
| Bilateral (vs right) | 2.05 | 0.65 to 6.45 | 0.221 |
| Reoperation | | | |
| Yes (vs no) | 1.13 | 0.35 to 3.70 | 0.836 |
| Midline shifting (mm) | | | |
| > 10 and ≤ 20 (vs ≤ 10) | 1.68 | 0.59 to 4.75 | 0.328 |
| > 20 (vs ≤ 10) | 6.20 | 0.70 to 54.61 | 0.100 |
| Initial Glasgow coma scale (per 1 score increase) | 0.88 | 0.71 to 1.10 | 0.273 |
| Traumatic SAH | | | |
| Yes (vs no) | 0.79 | 0.34 to 1.83 | 0.584 |
| Traumatic ICH | | | |
| Yes (vs no) | 1.04 | 0.48 to 2.27 | 0.924 |
| Traumatic IVH | | | |
| Yes (vs no) | 0.92 | 0.35 to 2.48 | 0.876 |
| Traumatic EDH | | | |
| Yes (vs no) | 1.02 | 0.38 to 2.76 | 0.962 |
| Skull fracture | | | |
| Yes (vs no) | 0.74 | 0.31 to 1.78 | 0.500 |
| Hypertension | 0.81 | 0.34 to 1.89 | 0.617 |
| Diabetes | 0.42 | 0.12 to 1.47 | 0.174 |

OR, odds ratio; CI, confidence interval; BMI, body mass index; HU, Hounsfield unit; IOP, internal occipital protuberance; SAH, subarachnoid hemorrhage; ICH, intracerebral hemorrhage; IVH, intraventricular hemorrhage; EDH, epidural hematoma

sensitivity = 78.8%; specificity = 58.1%; P<0.001), respectively (Fig 3C and 3D). According to the cut-off values, the study patients were then classified into (1) mean frontal skull HU ≤797.4, (2) IOP HU ≤586.5, and (3) simultaneous frontal skull HU ≤797.4 and IOP HU ≤586.5 groups.

## Independent predictive factors for SDHC

We performed univariate logistic regression analysis to screen possible factors associated with SDHC in study patients (Table 3).

Highest OR was observed in patients with simultaneous frontal skull HU ≤797.4 and IOP HU ≤586.5 (OR = 6.25) when compared to patients with mean frontal skull HU ≤797.4

(OR = 5.68) and IOP HU ≤586.5 (OR = 5.16) in univariate analysis. We thus selected the group with simultaneous frontal skull HU ≤797.4 and IOP HU ≤586.5 as the covariate in multivariable analysis. Multivariable logistic analysis revealed that the group with simultaneous frontal skull HU ≤797.4 and IOP HU ≤586.5 (OR, 8.57; 95% CI, 3.05 to 24.10; P<0.001) was the only independent predictor of SDHC in patients receiving cranioplasty after DC for traumatic acute SDH (Fig 4).

This multivariable logistic model was considered well-adjusted according to the Hosmer-Lemeshow test (P = 0.806, $\chi^2$ = 4.538). Unexpectedly, although a negative linear trend between age and HU values was observed in patients aged ≥55 years, there was no overall significant correlation between age and possible BMD in the study patients (S1 Fig). However, because age is strongly related to BMD, we performed an additional interaction analysis of SDHC between skull and IOP HU values and age. However, we observed that there were no interactions of SDHC between the mean frontal skull HU, IOP HU, and simultaneous skull HU and IOP HU binary groups and age (P = 0.507; P = 0.284; P = 0.721, respectively) (Fig 5).

In addition, we also tested the possible interaction between skull and IOP HU values and sex for SDHC development. Again, there were no interactions with respect to SDHC between the mean frontal skull HU and IOP HU and sex (P = 0.515; P = 0.397, respectively) (S2 Fig). Furthermore, we performed two separate multivariable logistic regressions using the same covariates as above with HU values as continuous covariates (S1 and S2 Tables). Both mean frontal skull and IOP HU values that were entered as continuous covariates into the multivariable logistic model were significant independent predictors of SDHC in patients receiving cranioplasty after DC for traumatic acute SDH (OR, 0.994; P<0.001; OR, 0.994; P<0.001, respectively).

## Discussion

The cut-off HU values of frontal skull and IOP for predicting SDHC were determined by ROC curve analysis. Based on the cut-off HU values of the skull, we identified that possible lower BMD group (simultaneous frontal skull HU ≤797.4 and IOP HU ≤586.5) demonstrated an approximately 8.6-fold increased risk of SDHC when compared with other patient groups receiving cranioplasty after DC for traumatic acute SDH after adjusting for other predictive factors. Therefore, we conjecture that HU measurement in cancellous bone of the skull may be predictive of SDHC development after DC for traumatic acute SDH. To the best of our knowledge, this study is the first to suggest a possible relationship between possible lower BMD and SDHC after DC for traumatic acute SDH.

Previous studies reported that specific regional cancellous bone HU values from CT scans showed strong correlation with T-score and may be useful for predicting osteoporotic conditions [16–19]. We previously demonstrated a strong correlation between T-score and frontal skull HU value [14,20]. As osteoporosis is a systemic disease, we propose that osteoporotic conditions may influence cancellous bone structures in the skull. In addition, osteoporosis is strongly associated with genetic components of type 1 collagen such as *COL1A1* and *COL1A2* [21]. A previous meta-analysis reported that polymorphism in the *COL1A1* gene is associated with a modest reduction in BMD and significant increase in risk of osteoporotic fractures [22].

Type 1 collagen is a major component of bone and a previous study reported that both the arachnoid trabeculae and granulations are also composed of type 1 collagen [8]. The subarachnoid trabeculae are delicate thin columns composed of abundant strands of collagen tissue that connect the arachnoid and pial membranes, and bring stability to the subarachnoid space and cerebrospinal fluid (CSF) flow [23]. We previously suggested that systemic osteoporosis may negatively affect the integrity of arachnoid trabeculae and granulation because bone,

arachnoid trabeculae, and granulations are composed of type 1 collagen [14,24]. Supporting our postulation, osteogenesis imperfecta, caused by heterozygous mutations in type 1 procollagen genes (*COL1A1/COL1A2*), is associated with communicating hydrocephalus [25].

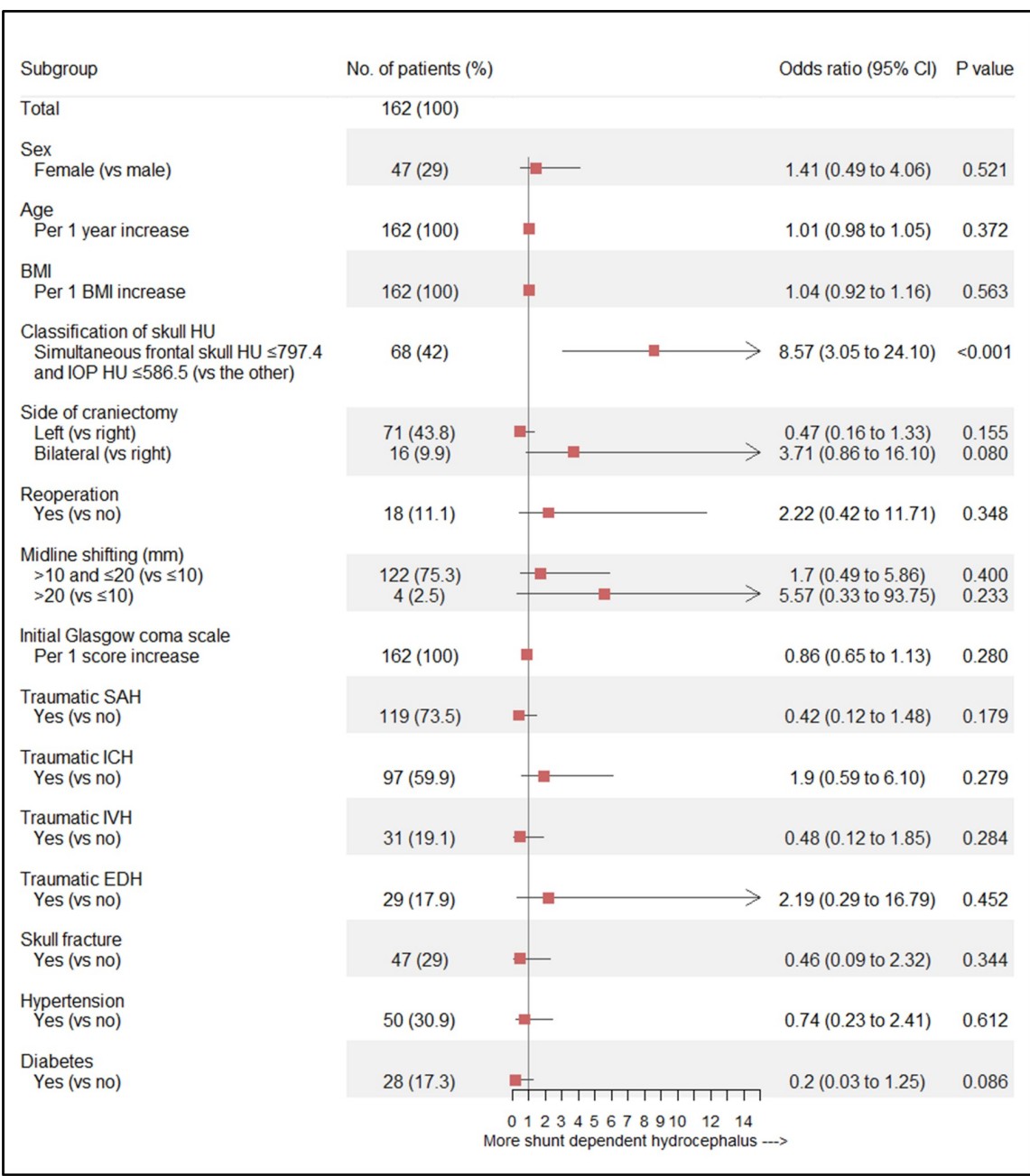

Hosmer-Lemeshow goodness of fit test: χ²=4.538, P=0.806

**Fig 4. Forest plot of estimates from multivariable logistic regression analysis for prediction of SDHC development according to potential predictive factors (adjusted for sex, age [continuous variable], BMI [continuous variable], group with simultaneous frontal skull HU ≤797.4 and IOP HU ≤586.5, side of craniectomy, reoperation, midline shifting, initial GCS [continuous variable], traumatic SAH, ICH, IVH, EDH, skull fracture, hypertension, and diabetes).** SDHC = shunt-dependent hydrocephalus; BMI = body mass index; HU = Hounsfield unit; IOP = internal occipital protuberance; GCS = Glasgow coma scale; SAH = subarachnoid hemorrhage; ICH = intracerebral hemorrhage; IVH = intraventricular hemorrhage; EDH = epidural hematoma; CI = confidence interval.

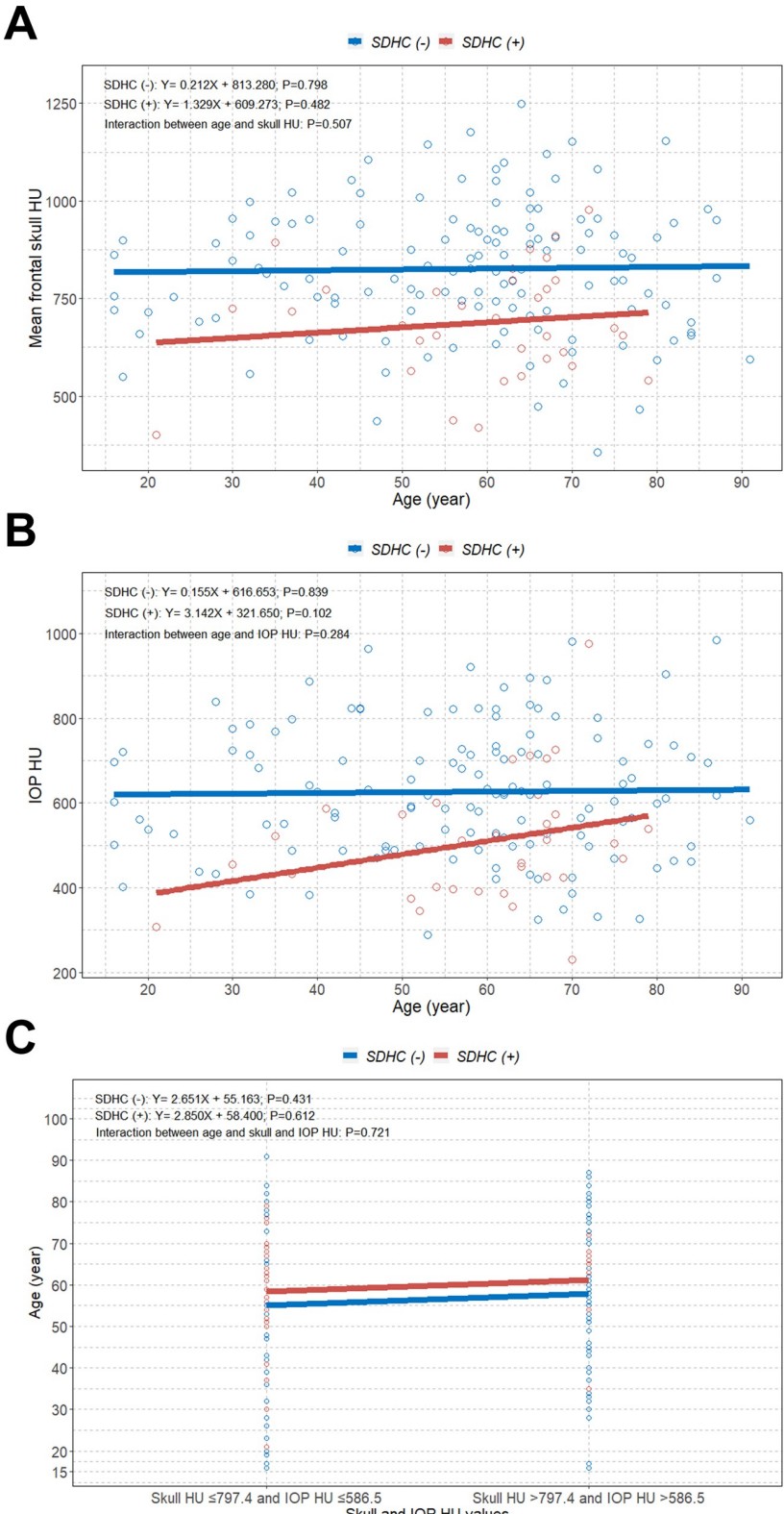

**Fig 5. Interaction plots between skull and IOP HU values and age of SDHC.** (A) between mean frontal skull HU and age. (B) between IOP HU and age. (C) between simultaneous skull HU and IOP HU binary groups and age. IOP = internal occipital protuberance; HU = Hounsfield unit; SDHC = shunt-dependent hydrocephalus.

Linear and angular acceleration of the brain due to sudden head trauma causes relative motion between structures of the brain and skull. This increases normal and shear stresses in the brain interface comprising CSF and subarachnoid space, including arachnoid trabeculae [26]. The complex consisting of CSF, arachnoid trabeculae, and subarachnoid vasculature, serves a critical role in protecting the brain from insults by acting as a mechanical cushion between the brain and skull, and absorbing energy upon impact [27].

Therefore, based on the findings and hypothesis, we suggest the following hypothetical mechanism as an explanation for the association between possible lower BMD and SDHC after DC for acute SDH. When head trauma occurs, normal and shear stresses may cause damage to the subarachnoid space including arachnoid trabeculae and granulations. After traumatic SDH occurs, mass effects of hematoma on the arachnoid surface of the brain with midline shifting may aggravate and accumulate damage to arachnoid trabeculae and granulations. Large cranial defects after DC may lead to turbulence in hydrodynamic CSF circulation and cerebral blood perfusion by atmospheric pressure, leading to the development of posttraumatic hydrocephalus [28,29]. Considering these plausible mechanisms, we hypothesized that the integrity of arachnoid trabeculae and granulations in the subarachnoid space after DC for SDH may be better maintained in those with denser, stronger type 1 collagen tissue than in patients with weaker type 1 collagen tissue. Maintenance of the integrity of arachnoid trabeculae and granulations may facilitate CSF resorption more efficiently by maintaining stability of the subarachnoid space and CSF flow through this structure when compared to that in patients with damaged or weaker arachnoid trabeculae and granulations.

Other possible predictive factors have been reported to be associated with posttraumatic hydrocephalus including intracranial pressure prior to decompression, subdural hygroma, low initial GCS score, increasing age, traumatic SAH, and traumatic IVH [30,31]. However, a previous study reported that interhemispheric hygromas after DC for severe head injury was the only independent prognostic factor for development of posttraumatic hydrocephalus [32].

Although there was a negative linear trend between age and HU value in patients aged ≥55 years, there was no overall correlation between age and HU value in the study patients. A previous study from Korea reported that the prevalence of osteoporosis was 8.8% and 39.1% in men and women aged 50 years or higher (50 to 89 years), respectively [33]. We think that the discrepancy of the association between age and possible BMD in our study could be because our study included only trauma patients and relatively younger patients (mean age, 57 years), with women only accounting for 29% of the study population. The interaction analysis showed that there was no significant effect of age on the HU value of SDHC development in the study patients.

Our study has several limitations. First, due to the retrospective nature of the study, our findings may be less accurate than those of a planned prospective study. Second, selection bias may have occurred, as we only included patients who ultimately underwent cranioplasty after DC for traumatic acute SDH in this study. In this regard, we did not include data from acute SDH patients who had undergone DC and died or were lost to follow-up due to transfer to other local hospitals for rehabilitation or other purposes. In addition, according to our hospital policy, permanent shunt insertion should be performed simultaneously with or after cranioplasty in craniectomy patients. Therefore, we inevitably limited patient populations in the study to reduce heterogeneity of data. Our study populations met narrow criteria that could have biased the results. Third, HU measurement errors may have occurred. However, all brain CT images were magnified for HU measurement, and we initially excluded patients with no measurable cancellous bone of the frontal skull in the brain CT scan.

In conclusion, our study suggests a relationship between possible low BMD and development of SDHC in traumatic acute SDH patients who had undergone DC. Our findings may

have value for predicting SDHC after DC for traumatic SDH using a convenient method for measuring HU using admission brain CT. The findings of this study may deepen our understanding of the association between low BMD and hydrocephalus after DC for traumatic acute SDH.

## Supporting information

**S1 Fig. Scatterplot with linear regression line showing the association between age and mean frontal skull and IOP HU values.** (A) association between age and mean frontal skull HU in all patients. (B) association between age and mean frontal skull HU in patients over 55 years of age. (C) association between age and IOP HU in all patients. (D) association between age and IOP HU in patients over 55 years of age. IOP = internal occipital protuberance; HU = Hounsfield unit.
(TIF)

**S2 Fig. Comparisons of mean frontal skull and IOP HU values between sex and interaction plots between skull and IOP HU values and sex of SDHC.** (A) boxplots with dot plots of mean frontal skull HU values according to the sex. (B) interaction between mean frontal skull HU and sex. (C) boxplots with dot plots of IOP HU values according to the sex. (D) interaction between IOP HU and sex. IOP = internal occipital protuberance; HU = Hounsfield unit; SDHC = shunt-dependent hydrocephalus.
(TIF)

**S1 Table. Multivariable logistic regression analysis for prediction of SDHC development according to potential predictive factors (adjusted for sex, age [continuous variable], BMI [continuous variable], mean frontal skull HU value [continuous variable], side of craniectomy, reoperation, midline shifting, initial GCS [continuous variable], traumatic SAH, ICH, IVH, EDH, skull fracture, hypertension, and diabetes).** SDHC = shunt-dependent hydrocephalus; BMI = body mass index; HU = Hounsfield unit; IOP = internal occipital protuberance; GCS = Glasgow coma scale; SAH = subarachnoid hemorrhage; ICH = intracerebral hemorrhage; IVH = intraventricular hemorrhage; EDH = epidural hematoma; CI = confidence interval.
(DOCX)

**S2 Table. Multivariable logistic regression analysis for prediction of SDHC development according to potential predictive factors (adjusted for sex, age [continuous variable], BMI [continuous variable], IOP HU value [continuous variable], side of craniectomy, reoperation, midline shifting, initial GCS [continuous variable], traumatic SAH, ICH, IVH, EDH, skull fracture, hypertension, and diabetes).** SDHC = shunt-dependent hydrocephalus; BMI = body mass index; HU = Hounsfield unit; IOP = internal occipital protuberance; GCS = Glasgow coma scale; SAH = subarachnoid hemorrhage; ICH = intracerebral hemorrhage; IVH = intraventricular hemorrhage; EDH = epidural hematoma; CI = confidence interval.
(DOCX)

## Author Contributions

**Conceptualization:** Myung-Hoon Han.

**Data curation:** In-Suk Bae.

**Formal analysis:** Myung-Hoon Han.

**Investigation:** In-Suk Bae.

**Methodology:** In-Suk Bae, Myung-Hoon Han.

**Resources:** Jin Hwan Cheong.

**Supervision:** Jae Min Kim, Jin Hwan Cheong, Je Il Ryu, Kyu-Sun Choi.

**Visualization:** Myung-Hoon Han.

**Writing – original draft:** In-Suk Bae, Myung-Hoon Han.

**Writing – review & editing:** Je Il Ryu.

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
