## [Decision Letter · Decision Letter 0]

11 Apr 2020

PONE-D-20-06156

Does the skull Hounsfield unit predict shunt dependent hydrocephalus after decompressive craniectomy for traumatic acute subdural hematoma?

PLOS ONE

Dear Prof Han,

Thank you for submitting your manuscript to PLOS ONE. After careful consideration, we feel that it has merit but does not fully meet PLOS ONE’s publication criteria as it currently stands. Therefore, we invite you to submit a revised version of the manuscript that addresses the points raised during the review process.

Please address the issues discussed by the reviewers, in particular the concerns about the statistics brought forward by reviewers #2 and #4, and change your manuscript accordingly.

We would appreciate receiving your revised manuscript by May 26 2020 11:59PM. To enhance the reproducibility of your results, we recommend that if applicable you deposit your laboratory protocols in protocols.io, where a protocol can be assigned its own identifier (DOI) such that it can be cited independently in the future. For instructions see: http://journals.plos.org/plosone/s/submission-guidelines#loc-laboratory-protocols

We look forward to receiving your revised manuscript.

Kind regards,

Michael C Burger, M.D.

Academic Editor

PLOS ONE

Journal Requirements:

2. Please note that all PLOS journals ask authors to adhere to our policies for sharing of data and materials: https://journals.plos.org/plosone/s/data-availability. According to PLOS ONE’s Data Availability policy, we require that the minimal dataset underlying results reported in the submission must be made immediately and freely available at the time of publication. As such, please remove any instances of 'unpublished data' or 'data not shown' in your manuscript and replace these with either the relevant data (in the form of additional figures, tables or descriptive text, as appropriate), a citation to where the data can be found, or remove altogether any statements supported by data not presented in the manuscript.

3. We noticed you have some minor occurrence(s) of overlapping text with the following previous publication(s), which needs to be addressed:

https://doi.org/10.1371/journal.pone.0226312

In your revision ensure you cite all your sources (including your own works), and quote or rephrase any duplicated text outside the Methods section. Further consideration is dependent on these concerns being addressed.

5. Thank you for including your ethics statement:  "This study was approved by the Institutional Review Boards of both hospitals and conformed with the tenets of the Declaration of Helsinki. Owing to the retrospective nature of the study, the requirement for informed consent was waived. All individual records were anonymized prior to analysis."

Reviewers' comments:

Reviewer's Responses to Questions

**Comments to the Author**

1. Is the manuscript technically sound, and do the data support the conclusions?

Reviewer #1: Yes

Reviewer #2: Partly

Reviewer #3: Yes

Reviewer #4: Yes

2. Has the statistical analysis been performed appropriately and rigorously? 

Reviewer #1: Yes

Reviewer #2: No

Reviewer #3: Yes

Reviewer #4: I Don't Know

3. Have the authors made all data underlying the findings in their manuscript fully available?

Reviewer #1: Yes

Reviewer #2: No

Reviewer #3: Yes

Reviewer #4: Yes

4. Is the manuscript presented in an intelligible fashion and written in standard English?

Reviewer #1: Yes

Reviewer #2: Yes

Reviewer #3: Yes

Reviewer #4: Yes

5. Review Comments to the Author

Reviewer #1: I was very sceptic after reading the title of this manuscript, but became more and more enthusiastic when reading the paper. The authors have tested out a very novel hypothesis: Can skull mineral bone density predict the need for VP-shunt after decompressive craniectomy (DC) for acute SDH? The theoretical basis for the hypothesis is that collagen type 1 is a major component of bone and subarachnoid trabeculae/granulations, and that dysregulation of collagen typ1 formation may be associated with both osteoporosis and degeneration of arachnoid trabeculae/granulations. Weakening of arachnoid trabeculae/granulations may in turn predispose to enlarged subarachnoid space/hydrocephalus. This is a combination great knowledge and originality.

The authors present good evidence that low density of the cancellous bone of the skull may be predictive of VP-shunt placement after DC for traumatic acute SDH.

Patient sample, methodology and results are well presented. The manuscript is easy to read and the Language is good.

The limitations of the study are well described in the Discussion.

I have no major objections to this fascinating study, and I hope other groups will repeat this study.

Reviewer #2: The authors provide a remarkable and fascinating correlation between bone mineral density and the likelihood of post-craniectomy hydrocephalus. If true, this is very important. However, there are statistical issues which dampen my enthusiasm and are highlighted below. Though I’m optimistic that the results will hold with more rigorous stats.

The statistical methodology has two significant flaws. The first has to do with converting a numerical variable (HU) to a categorical one. The authors use ROC analysis to choose a threshold that optimally segregates hydro from non-hydro patients. They then use that threshold in the multivariable model. Since they’ve already proven this threshold significantly separates the two classes, it’s not surprising that the multivariable model finds the same thing. Why not leave HU as a numeric variable or ordinal variable, rather than bifurcating it? This was done with other numeric variables like age and BMI, so clearly the authors can do it. I am hopeful that this supplemental analysis (leaving HU numerical) won’t change the overall outcome of the study, but the authors should do it to reassure skeptical readers.

The second problem is that there is no effort to test the fit of their multivariable model. It has a ton of variables and there’s no evidence they did anything to reduce the model. Having too many variables can easily mask significant interactions. An example of how to do this in R is given in this publication: https://www.ncbi.nlm.nih.gov/pmc/articles/PMC4828741/ . The authors need to convince us that there is no significant interaction among variables in the model they’ve created, and that we’re not missing an explanatory co-factor (like age correlating with HU, which it certainly does in all my patients).

There is no mention of the data sharing for this paper, or that the data is publicly available, which is required for PLoS ONE submissions. This needs to be explicitly described and provided.

Minor issues:

The authors presumably mean “multivariable” regression and not “multivariate” regression. This is a common mistake. See https://www.ncbi.nlm.nih.gov/pmc/articles/PMC3518362/

Was there any missing data for any patient in the model? The paper should explicitly state that there was not, if there was not. If there WAS, they should absolutely document that.

More detail must be given about how patients were selected for shunts. The authors say “hydrocephalus symptoms”, which is far too vague.

Reviewer #3: The authors tested a hypothesis regarding calvarial bone density and post traumatic hydrocephalus in an appropriately sized, well defined patient population. Their hypothesis (decreased bone density correlating with post traumatic hydrocephalus in a subset of patients who undergo decompressive craniectomy) is novel and potentially useful for practicing neurosurgeons. The study is well designed and clearly presented.

Reviewer #4: This is a well written manuscript investigating the relationship of bone mineral density as determined by skull hounsfield units and shunt dependent hydrocephalus. There were 162 patients included, but only 33 developed SDHC.

- it is interesting that there is significant correlation with skull HU and SDHC, but not age. One would expect that decreased skull HU would be very much age-dependent (as well as sex-dependent). This may be due to the fact that age odds ratios were measured per 1-year increase, while HU were measured in a binary fashion. The authors should discuss in the discussion why this expected relationship does not exist.

- "shunt dependent" is difficult to assess, especially in post-traumatic hydrocephalus as sometimes the indications for shunt placement vary by practitioner. The authors should add additional information on how they determined the need for shunt placement, and what they consider to be hydrocephalus-related symptoms (ie, depressed level of consciousness only, or subtler symptoms like headache).

Overall, I believe this is a valuable addition to the literature

6. PLOS authors have the option to publish the peer review history of their article (what does this mean?). If published, this will include your full peer review and any attached files.

Reviewer #1: Yes: Eirik Helseth

Reviewer #2: No

Reviewer #3: Yes: Ronald Benveniste MD

Reviewer #4: Yes: Jacob R. Joseph, MD

---

## [Author Response · Author response to Decision Letter 0]

16 Apr 2020

Response to reviewers 

Does the skull Hounsfield unit predict shunt dependent hydrocephalus after decompressive craniectomy for traumatic acute subdural hematoma?

In-Suk Bae, Jae Min Kim, Jin Hwan Cheong, Je Il Ryu, Kyu-Sun Choi, Myung-Hoon Han

Reviewer #2 

The statistical methodology has two significant flaws. The first has to do with converting a numerical variable (HU) to a categorical one. The authors use ROC analysis to choose a threshold that optimally segregates hydro from non-hydro patients. They then use that threshold in the multivariable model. Since they’ve already proven this threshold significantly separates the two classes, it’s not surprising that the multivariable model finds the same thing. Why not leave HU as a numeric variable or ordinal variable, rather than bifurcating it? This was done with other numeric variables like age and BMI, so clearly the authors can do it. I am hopeful that this supplemental analysis (leaving HU numerical) won’t change the overall outcome of the study, but the authors should do it to reassure skeptical readers.

Response: Thank you for your comment and we agree with you. We performed two separate multivariable logistic regressions with HU values as continuous covariates and presented the Supplemental Tables as you suggested as follows. 

S1 Table. Multivariable logistic regression analysis for prediction of SDHC development according to potential predictive factors (adjusted for sex, age [continuous variable], BMI [continuous variable], mean frontal skull HU value [continuous variable], side of craniectomy, reoperation, midline shifting, initial GCS [continuous variable], traumatic SAH, ICH, IVH, EDH, skull fracture, hypertension, and diabetes). SDHC=shunt-dependent hydrocephalus; BMI=body mass index; HU=Hounsfield unit; IOP=internal occipital protuberance; GCS= Glasgow coma scale; SAH=subarachnoid hemorrhage; ICH=intracerebral hemorrhage; IVH=intraventricular hemorrhage; EDH=epidural hematoma; CI=confidence interval.

 Multivariable logistic regression analysis

Variable OR 95%CI P

Sex 

Female (vs male) 1.260 0.439 to 3.618 0.668

Age (per 1-year increase) 1.016 0.985 to 1.047 0.311

BMI (per 1 BMI increase) 1.043 0.927 to 1.175 0.482

Mean frontal skull HU (per 1 HU increase) 0.994 0.991 to 0.997 <0.001

Side of craniectomy 

Left (vs right) 0.467 0.165 to 1.326 0.153

Bilateral (vs right) 4.055 0.955 to 17.217 0.058

Reoperation 

Yes (vs no) 1.620 0.348 to 7.538 0.538

Midline shifting (mm) 

> 10 and ≤ 20 (vs ≤ 10) 1.636 0.461 to 5.810 0.447

> 20 (vs ≤ 10) 8.224 0.553 to 122.391 0.126

Initial Glasgow coma scale 

(per 1 score increase) 0.843 0.639 to 1.113 0.228

Traumatic SAH 

Yes (vs no) 0.475 0.141 to 1.603 0.230

Traumatic ICH 

Yes (vs no) 1.400 0.446 to 4.394 0.564

Traumatic IVH 

Yes (vs no) 0.560 0.144 to 2.169 0.401

Traumatic EDH 

Yes (vs no) 2.204 0.295 to 16.478 0.441

Skull fracture 

Yes (vs no) 0.546 0.104 to 2.871 0.475

Hypertension 0.576 0.175 to 1.896 0.364

Diabetes 0.261 0.047 to 1.431 0.122

Hosmer-Lemeshow goodness of fit test: χ2=12.968, P=0.113.

OR, odds ratio; CI, confidence interval; BMI, body mass index; HU, Hounsfield unit; SAH, subarachnoid hemorrhage; ICH, intracerebral hemorrhage; IVH, intraventricular hemorrhage; EDH, epidural hematoma.

S2 Table. Multivariable logistic regression analysis for prediction of SDHC development according to potential predictive factors (adjusted for sex, age [continuous variable], BMI [continuous variable], IOP HU value [continuous variable], side of craniectomy, reoperation, midline shifting, initial GCS [continuous variable], traumatic SAH, ICH, IVH, EDH, skull fracture, hypertension, and diabetes). SDHC=shunt-dependent hydrocephalus; BMI=body mass index; HU=Hounsfield unit; IOP=internal occipital protuberance; GCS= Glasgow coma scale; SAH=subarachnoid hemorrhage; ICH=intracerebral hemorrhage; IVH=intraventricular hemorrhage; EDH=epidural hematoma; CI=confidence interval.

 Multivariable logistic regression analysis

Variable OR 95%CI P

Sex 

Female (vs male) 1.528 0.531 to 4.394 0.431

Age (per 1-year increase) 1.019 0.987 to 1.052 0.239

BMI (per 1 BMI increase) 1.038 0.923 to 1.167 0.533

IOP HU (per 1 HU increase) 0.993 0.989 to 0.997 <0.001

Side of craniectomy 

Left (vs right) 0.473 0.168 to 1.332 0.157

Bilateral (vs right) 4.934 1.139 to 21.373 0.033

Reoperation 

Yes (vs no) 1.813 0.364 to 9.032 0.468

Midline shifting (mm) 

> 10 and ≤ 20 (vs ≤ 10) 1.839 0.532 to 6.354 0.336

> 20 (vs ≤ 10) 6.280 0.400 to 98.641 0.191

Initial Glasgow coma scale 

(per 1 score increase) 0.869 0.658 to 1.148 0.324

Traumatic SAH 

Yes (vs no) 0.505 0.152 to 1.681 0.266

Traumatic ICH 

Yes (vs no) 1.633 0.515 to 5.175 0.404

Traumatic IVH 

Yes (vs no) 0.564 0.144 to 2.202 0.409

Traumatic EDH 

Yes (vs no) 3.413 0.430 to 27.103 0.246

Skull fracture 

Yes (vs no) 0.341 0.062 to 1.871 0.216

Hypertension 0.642 0.196 to 2.103 0.464

Diabetes 0.245 0.044 to 1.352 0.106

Hosmer-Lemeshow goodness of fit test: χ2=9.783, P=0.281.

OR, odds ratio; CI, confidence interval; BMI, body mass index; IOP, internal occipital protuberance; HU, Hounsfield unit; SAH, subarachnoid hemorrhage; ICH, intracerebral hemorrhage; IVH, intraventricular hemorrhage; EDH, epidural hematoma.

We have added relevant sentences to the “Independent predictive factors for SDHC” section of the Results as follows. 

Results

Independent predictive factors for SDHC 

Furthermore, we performed two separate multivariable logistic regressions using the same covariates as above with HU values as continuous covariates (S1 and S2 Tables). Both mean frontal skull and IOP HU values that were entered as continuous covariates into the multivariable logistic model were significant independent predictors of SDHC in patients receiving cranioplasty after DC for traumatic acute SDH (OR, 0.994; P<0.001; OR, 0.994; P<0.001, respectively).

The second problem is that there is no effort to test the fit of their multivariable model. It has a ton of variables and there’s no evidence they did anything to reduce the model. Having too many variables can easily mask significant interactions. An example of how to do this in R is given in this publication: https://www.ncbi.nlm.nih.gov/pmc/articles/PMC4828741/ . The authors need to convince us that there is no significant interaction among variables in the model they’ve created, and that we’re not missing an explanatory co-factor (like age correlating with HU, which it certainly does in all my patients).

Response: Thank you for your comment and we agree with you. We performed the Hosmer-Lemeshow goodness of fit test on the logistic regression model as you suggested. Data seen below: 

We have added the results of the Hosmer-Lemeshow goodness of fit test to Figure 4 as follows.

We also performed a linear regression analysis to assess the association between age and HU value. We present the results in S1 Fig. Unexpectedly, although a negative linear trend between age and HU values was observed in patients aged ≥55 years, there was no overall significant correlation between age and possible BMD in the study patients, as seen below. 

A previous study from Korea reported that the prevalence of osteoporosis was 8.8% and 39.1% in men and women aged 50 years or higher (50 to 89 years), respectively (Lee J, Lee S, Jang S, Ryu OH. Age-Related Changes in the Prevalence of Osteoporosis according to Gender and Skeletal Site: The Korea National Health and Nutrition Examination Survey 2008-2010. Endocrinol Metab. 2013;28: 180–191). We think that the discrepancy of association between age and possible BMD in our study could be because our study included only trauma patients and relatively younger patients (mean age, 57 years) with women accounting for 29% of the study population. The interaction analysis showed that there was no significant effect of age on the HU value of SDHC development in the study patients. 

We also added interaction plots and analyses that were performed using R software as follows.

Figure 5

S2 Fig.

We have also added relevant sentences to the Methods, Results, and Discussion sections as follows.

Methods

Statistical methods

Odds ratios (ORs) with 95% confidence intervals (CIs) were estimated using uni- and multivariate logistic regressions to determine the independent predictive factors for the development of SDHC in patients that received cranioplasty after DC for acute SDH. Sex, age (continuous variable), BMI (continuous variable), patients with simultaneous frontal skull HU ≤797.4 and IOP HU ≤586.5, side of craniectomy, reoperation, midline shifting, initial GCS (continuous variable), traumatic SAH, ICH, IVH, EDH, skull fracture, hypertension, and diabetes were entered into the multivariate model. The Hosmer-Lemeshow goodness of fit test was used to analyze the fit of the logistic regression model. Additional interaction analysis was performed using the R package, “effects”.

P-values less than 0.05 were considered statistically significant. All statistical analyses were performed using R software version 3.5.2 and SPSS for Windows, version 24.0 software (IBM, Chicago, IL).

Results

Independent predictive factors for SDHC 

This multivariable logistic model was considered well-adjusted according to the Hosmer-Lemeshow test (P=0.806, χ2=4.538). Unexpectedly, although a negative linear trend between age and HU values was observed in patients aged ≥55 years, there was no overall significant correlation between age and possible BMD in the study patients (S1 Fig). However, because age is strongly related to BMD, we performed an additional interaction analysis of SDHC between skull and IOP HU values and age. However, we observed that there were no interactions of SDHC between the mean frontal skull HU, IOP HU, and simultaneous skull HU and IOP HU binary groups and age (P=0.507; P=0.284; P=0.721, respectively) (Fig. 5). 

Figure 5. Interaction plots between skull and IOP HU values and age of SDHC. (A) between mean frontal skull HU and age. (B) between IOP HU and age. (C) between simultaneous skull HU and IOP HU binary groups and age. IOP=internal occipital protuberance; HU=Hounsfield unit; SDHC=shunt-dependent hydrocephalus.

In addition, we also tested the possible interaction between skull and IOP HU values and sex for SDHC development. Again, there were no interactions with respect to SDHC between the mean frontal skull HU and IOP HU and sex (P=0.515; P=0.397, respectively) (S2 Fig).

Discussion

Other possible predictive factors have been reported to be associated with posttraumatic hydrocephalus including intracranial pressure prior to decompression, subdural hygroma, low initial GCS score, increasing age, traumatic SAH, and traumatic IVH [31,31]. However, a previous study reported that interhemispheric hygromas after DC for severe head injury was the only independent prognostic factor for development of posttraumatic hydrocephalus [32].

Although there was a negative linear trend between age and HU value in patients aged ≥55 years, there was no overall correlation between age and HU value in the study patients. A previous study from Korea reported that the prevalence of osteoporosis was 8.8% and 39.1% in men and women aged 50 years or higher (50 to 89 years), respectively [33]. We think that the discrepancy of the association between age and possible BMD in our study could be because our study included only trauma patients and relatively younger patients (mean age, 57 years), with women only accounting for 29% of the study population. The interaction analysis showed that there was no significant effect of age on the HU value of SDHC development in the study patients. 

Our study has several limitations. 

There is no mention of the data sharing for this paper, or that the data is publicly available, which is required for PLoS ONE submissions. This needs to be explicitly described and provided. 

Response: We added a data availability statement to the Methods section as follows. 

Methods

Data Availability Statement: All relevant data are within the manuscript and its Supporting Information files.

Minor issues:

The authors presumably mean “multivariable” regression and not “multivariate” regression. This is a common mistake. See https://www.ncbi.nlm.nih.gov/pmc/articles/PMC3518362/

Response: Thank you for your comment and we apologize for our error. We have changed all expressions of “multivariate” with “multivariable” in the revised manuscript, as you suggested.

Was there any missing data for any patient in the model? The paper should explicitly state that there was not, if there was not. If there WAS, they should absolutely document that.

Response: Thank you for your comment. There were no missing data for any patient because all study patients were well followed up after decompressive craniectomy and underwent subsequent cranioplasty. We added a clarifying sentence to the Methods section as follows.

Methods

Study design

The remaining 162 patients were finally included in the study. All study patients were well followed up after DC and underwent subsequent cranioplasty; there were no missing data.

More detail must be given about how patients were selected for shunts. The authors say “hydrocephalus symptoms”, which is far too vague.

Response: Thank you for your comment and we agree. We have added relevant sentences to the Surgery and management section of the Methods as follows.

Methods

Surgery and management 

V-P shunt was performed when the patients had hydrocephalus-related symptoms with the presence of ventricular enlargement on brain CT scans. Hydrocephalus-related symptoms for considering V-P shunt were defined as neurological deterioration and aggravated response to painful stimuli, especially in stuporous patients, and additional decline in the daily functions, gait disturbance, urinary incontinence, or confusion, especially in conscious patients. Simultaneous or staged V-P shunting was performed with or following cranioplasty.

Reviewer #4 

- it is interesting that there is significant correlation with skull HU and SDHC, but not age. One would expect that decreased skull HU would be very much age-dependent (as well as sex-dependent). This may be due to the fact that age odds ratios were measured per 1-year increase, while HU were measured in a binary fashion. The authors should discuss in the discussion why this expected relationship does not exist.

Response: Thank you for your comment and we agree with you. We performed two separate multivariable logistic regressions with HU values as continuous covariates and presented the Supplemental Tables as you suggested as follows.

S1 Table. Multivariable logistic regression analysis for prediction of SDHC development according to potential predictive factors (adjusted for sex, age [continuous variable], BMI [continuous variable], mean frontal skull HU value [continuous variable], side of craniectomy, reoperation, midline shifting, initial GCS [continuous variable], traumatic SAH, ICH, IVH, EDH, skull fracture, hypertension, and diabetes). SDHC=shunt-dependent hydrocephalus; BMI=body mass index; HU=Hounsfield unit; IOP=internal occipital protuberance; GCS= Glasgow coma scale; SAH=subarachnoid hemorrhage; ICH=intracerebral hemorrhage; IVH=intraventricular hemorrhage; EDH=epidural hematoma; CI=confidence interval.

 Multivariable logistic regression analysis

Variable OR 95%CI P

Sex 

Female (vs male) 1.260 0.439 to 3.618 0.668

Age (per 1-year increase) 1.016 0.985 to 1.047 0.311

BMI (per 1 BMI increase) 1.043 0.927 to 1.175 0.482

Mean frontal skull HU (per 1 HU increase) 0.994 0.991 to 0.997 <0.001

Side of craniectomy 

Left (vs right) 0.467 0.165 to 1.326 0.153

Bilateral (vs right) 4.055 0.955 to 17.217 0.058

Reoperation 

Yes (vs no) 1.620 0.348 to 7.538 0.538

Midline shifting (mm) 

> 10 and ≤ 20 (vs ≤ 10) 1.636 0.461 to 5.810 0.447

> 20 (vs ≤ 10) 8.224 0.553 to 122.391 0.126

Initial Glasgow coma scale 

(per 1 score increase) 0.843 0.639 to 1.113 0.228

Traumatic SAH 

Yes (vs no) 0.475 0.141 to 1.603 0.230

Traumatic ICH 

Yes (vs no) 1.400 0.446 to 4.394 0.564

Traumatic IVH 

Yes (vs no) 0.560 0.144 to 2.169 0.401

Traumatic EDH 

Yes (vs no) 2.204 0.295 to 16.478 0.441

Skull fracture 

Yes (vs no) 0.546 0.104 to 2.871 0.475

Hypertension 0.576 0.175 to 1.896 0.364

Diabetes 0.261 0.047 to 1.431 0.122

Hosmer-Lemeshow goodness of fit test: χ2=12.968, P=0.113.

OR, odds ratio; CI, confidence interval; BMI, body mass index; HU, Hounsfield unit; SAH, subarachnoid hemorrhage; ICH, intracerebral hemorrhage; IVH, intraventricular hemorrhage; EDH, epidural hematoma.

S2 Table. Multivariable logistic regression analysis for prediction of SDHC development according to potential predictive factors (adjusted for sex, age [continuous variable], BMI [continuous variable], IOP HU value [continuous variable], side of craniectomy, reoperation, midline shifting, initial GCS [continuous variable], traumatic SAH, ICH, IVH, EDH, skull fracture, hypertension, and diabetes). SDHC=shunt-dependent hydrocephalus; BMI=body mass index; HU=Hounsfield unit; IOP=internal occipital protuberance; GCS= Glasgow coma scale; SAH=subarachnoid hemorrhage; ICH=intracerebral hemorrhage; IVH=intraventricular hemorrhage; EDH=epidural hematoma; CI=confidence interval.

 Multivariable logistic regression analysis

Variable OR 95%CI P

Sex 

Female (vs male) 1.528 0.531 to 4.394 0.431

Age (per 1-year increase) 1.019 0.987 to 1.052 0.239

BMI (per 1 BMI increase) 1.038 0.923 to 1.167 0.533

IOP HU (per 1 HU increase) 0.993 0.989 to 0.997 <0.001

Side of craniectomy 

Left (vs right) 0.473 0.168 to 1.332 0.157

Bilateral (vs right) 4.934 1.139 to 21.373 0.033

Reoperation 

Yes (vs no) 1.813 0.364 to 9.032 0.468

Midline shifting (mm) 

> 10 and ≤ 20 (vs ≤ 10) 1.839 0.532 to 6.354 0.336

> 20 (vs ≤ 10) 6.280 0.400 to 98.641 0.191

Initial Glasgow coma scale 

(per 1 score increase) 0.869 0.658 to 1.148 0.324

Traumatic SAH 

Yes (vs no) 0.505 0.152 to 1.681 0.266

Traumatic ICH 

Yes (vs no) 1.633 0.515 to 5.175 0.404

Traumatic IVH 

Yes (vs no) 0.564 0.144 to 2.202 0.409

Traumatic EDH 

Yes (vs no) 3.413 0.430 to 27.103 0.246

Skull fracture 

Yes (vs no) 0.341 0.062 to 1.871 0.216

Hypertension 0.642 0.196 to 2.103 0.464

Diabetes 0.245 0.044 to 1.352 0.106

Hosmer-Lemeshow goodness of fit test: χ2=9.783, P=0.281.

OR, odds ratio; CI, confidence interval; BMI, body mass index; IOP, internal occipital protuberance; HU, Hounsfield unit; SAH, subarachnoid hemorrhage; ICH, intracerebral hemorrhage; IVH, intraventricular hemorrhage; EDH, epidural hematoma.

We have added relevant sentences to the “Independent predictive factors for SDHC” section of the Results as follows. 

Results

Independent predictive factors for SDHC 

Furthermore, we performed two separate multivariable logistic regressions using the same covariates as above with HU values as continuous covariates (S1 and S2 Tables). Both mean frontal skull and IOP HU values that were entered as continuous covariates into the multivariable logistic model were significant independent predictors of SDHC in patients receiving cranioplasty after DC for traumatic acute SDH (OR, 0.994; P<0.001; OR, 0.994; P<0.001, respectively).

We also performed a linear regression analysis to assess the association between age and HU value. We present the results in S1 Fig. Unexpectedly, although a negative linear trend between age and HU values was observed in patients aged ≥55 years, there was no overall significant correlation between age and possible BMD in the study patients, as seen below.

A previous study from Korea reported that the prevalence of osteoporosis was 8.8% and 39.1% in men and women aged 50 years or higher (50 to 89 years), respectively (Lee J, Lee S, Jang S, Ryu OH. Age-Related Changes in the Prevalence of Osteoporosis according to Gender and Skeletal Site: The Korea National Health and Nutrition Examination Survey 2008-2010. Endocrinol Metab. 2013;28: 180–191). We think that the discrepancy of association between age and possible BMD in our study could be because our study included only trauma patients and relatively younger patients (mean age, 57 years) with women accounting for 29% of the study population. The interaction analysis showed that there was no significant effect of age on the HU value of SDHC development in the study patients. 

We also added interaction plots and analyses that were performed using R software as follows.

Figure 5

S2 Fig.

We have also added relevant sentences to the Methods, Results, and Discussion sections as follows.

Methods

Statistical methods

Odds ratios (ORs) with 95% confidence intervals (CIs) were estimated using uni- and multivariate logistic regressions to determine the independent predictive factors for the development of SDHC in patients that received cranioplasty after DC for acute SDH. Sex, age (continuous variable), BMI (continuous variable), patients with simultaneous frontal skull HU ≤797.4 and IOP HU ≤586.5, side of craniectomy, reoperation, midline shifting, initial GCS (continuous variable), traumatic SAH, ICH, IVH, EDH, skull fracture, hypertension, and diabetes were entered into the multivariate model. The Hosmer-Lemeshow goodness of fit test was used to analyze the fit of the logistic regression model. Additional interaction analysis was performed using the R package, “effects”.

P-values <0.05 were considered statistically significant. All statistical analyses were performed using R software version 3.5.2 and SPSS for Windows, version 24.0 software (IBM, Chicago, IL).

Results

Independent predictive factors for SDHC 

Unexpectedly, although a negative linear trend between age and HU values was observed in patients aged ≥55 years, there was no overall significant correlation between age and possible BMD in the study patients (S1 Fig). However, because age is strongly related to BMD, we performed an additional interaction analysis of SDHC between skull and IOP HU values and age. However, we observed that there were no interactions of SDHC between the mean frontal skull HU, IOP HU, and simultaneous skull HU and IOP HU binary groups and age (P=0.507; P=0.284; P=0.721, respectively) (Fig. 5). 

Figure 5. Interaction plots between skull and IOP HU values and age of SDHC. (A) between mean frontal skull HU and age. (B) between IOP HU and age. (C) between simultaneous skull HU and IOP HU binary groups and age. IOP=internal occipital protuberance; HU=Hounsfield unit; SDHC=shunt-dependent hydrocephalus.

In addition, we also tested the possible interaction between skull and IOP HU values and sex for SDHC development. Again, there were no interactions with respect to SDHC between the mean frontal skull HU and IOP HU and sex (P=0.515; P=0.397, respectively) (S2 Fig).

Discussion

Other possible predictive factors have been reported to be associated with posttraumatic hydrocephalus including intracranial pressure prior to decompression, subdural hygroma, low initial GCS score, increasing age, traumatic SAH, and traumatic IVH [31,31]. However, a previous study reported that interhemispheric hygromas after DC for severe head injury was the only independent prognostic factor for development of posttraumatic hydrocephalus [32].

Although there was a negative linear trend between age and HU value in patients aged ≥55 years, there was no overall correlation between age and HU value in the study patients. A previous study from Korea reported that the prevalence of osteoporosis was 8.8% and 39.1% in men and women aged 50 years or higher (50 to 89 years), respectively [33]. We think that the discrepancy of the association between age and possible BMD in our study could be because our study included only trauma patients and relatively younger patients (mean age, 57 years), with women only accounting for 29% of the study population. The interaction analysis showed that there was no significant effect of age on the HU value of SDHC development in the study patients. 

Our study has several limitations. 

- "shunt dependent" is difficult to assess, especially in post-traumatic hydrocephalus as sometimes the indications for shunt placement vary by practitioner. The authors should add additional information on how they determined the need for shunt placement, and what they consider to be hydrocephalus-related symptoms (ie, depressed level of consciousness only, or subtler symptoms like headache) 

Response: Thank you for your comment and we agree. We have added relevant sentences to the Surgery and management section of the Methods as follows.

Methods

Surgery and management 

V-P shunt was performed when the patients had hydrocephalus-related symptoms with the presence of ventricular enlargement on brain CT scans. Hydrocephalus-related symptoms for considering V-P shunt were defined as neurological deterioration and aggravated response to painful stimuli, especially in stuporous patients, and additional decline in the daily functions, gait disturbance, urinary incontinence, or confusion, especially in conscious patients. Simultaneous or staged V-P shunting was performed with or following cranioplasty.

---

## [Editor Report · Decision Letter 1]

20 Apr 2020

Does the skull Hounsfield unit predict shunt dependent hydrocephalus after decompressive craniectomy for traumatic acute subdural hematoma?

PONE-D-20-06156R1

Dear Dr. Han,

We are pleased to inform you that your manuscript has been judged scientifically suitable for publication and will be formally accepted for publication once it complies with all outstanding technical requirements. You have adressed all objections and suggestions for improvement brought forward by the reviewers convincingly.

With kind regards,

Michael C Burger, M.D.

Academic Editor

PLOS ONE

---

## [Editor Report · Acceptance letter]

22 Apr 2020

PONE-D-20-06156R1 

Does the skull Hounsfield unit predict shunt dependent hydrocephalus after decompressive craniectomy for traumatic acute subdural hematoma? 

Dear Dr. Han:

I am pleased to inform you that your manuscript has been deemed suitable for publication in PLOS ONE. Congratulations! Your manuscript is now with our production department. 

With kind regards,

on behalf of

Dr. Michael C Burger 

Academic Editor

PLOS ONE